# Behavioral and cultural determinants of symptomatic dry eye disease among university students in the UAE

Mona Aridi[1], Nisreen Alwan[2,3], Wissam Ghach[4]*

1 University of Angers, LARIS, SFR MATHSTIC, Angers, France, 2 College of Health Sciences, Abu Dhabi University, Abu Dhabi, United Arab Emirates, 3 Vision Care Association, Beirut, Lebanon, 4 Department of Public Health, Canadian University Dubai, Dubai, United Arab Emirates

* wissam.ghach@cud.ac.ae

## Abstract

### Objectives

This study aims to estimate the prevalence of symptomatic Dry Eye Disease (DED) and its associated risk factors among university students in the UAE.

### Methods

A cross-sectional study was conducted among 654 participants using the validated Ocular Surface Disease Index (OSDI) questionnaire, distributed via an online Google survey. The survey assessed the prevalence of DED along with demographic and behavioral-cultural risk factors. Kruskal-Wallis test was employed to analyze group differences. Spearman's rank correlation evaluated associations between OSDI scores and quantitative variables.

### Results

The study revealed significant gender differences, with females reporting higher OSDI scores than males. Smoking behaviors were strongly associated with OSDI scores, with daily smokers and those smoking in indoor environments. Among smoking types, Dokha and Mouassal waterpipe users exhibited the highest median OSDI scores. Eye cosmetic practices with users of shorter annual usage reported higher scores, while effective cleansing methods, such as soap and water or cleansing creams, were associated with lower scores. Prolonged screen time and extended study hours were positively correlated with OSDI scores.

### Conclusions

Symptomatic DED is highly prevalent among UAE university students. These findings underscore the need for targeted awareness and prevention strategies to mitigate DED risk factors within this population.

**Data availability statement:** All relevant data are within the manuscript and its Supporting Information files.

**Funding:** The author(s) received no specific funding for this work.

**Competing interests:** The authors have declared that no competing interests exist.

## 1. Introduction

Dry Eye Disease (DED) is a multifactorial condition of the ocular surface that disrupts tear film homeostasis and is accompanied by ocular symptoms. The Tear Film and Ocular Surface Society Dry Eye Workshop II (TFOS DEWS II) defines DED as "a loss of homeostasis of the tear film accompanied by tear film instability, hyperosmolarity, ocular surface inflammation, damage, and neurosensory abnormalities" [1]. The prevalence of DED is significant, ranging globally from 5% to 50%, with Arab countries reporting some of the highest rates [2] particularly, among the Jordanian population [3]. Likewise, the prevalence of DED in Middle Eastern countries is higher than the global estimate, highlighting a significant regional variation [4]. This variability is attributed to factors such as geography, climate, and diagnostic criteria, highlighting the importance of region-specific investigations.

DED prevalence is typically assessed using clinical tools such as Tear Break-Up Time (TBUT) and Schirmer Test, or symptom-based questionnaires, including the Ocular Surface Disease Index (OSDI) and Standard Patient Evaluation of Eye Dryness (SPEED) [5,6]. In addition to environmental and genetic factors, behavioral and cultural habits, including smoking and eye cosmetic use, have emerged as critical risk factors [3,7,8].

Smoking remains a significant global public health issue, with tobacco products containing over 4,000 toxic chemicals that contribute to cancers, cardiovascular diseases, and ocular conditions [9]. Smoking has been shown to destabilize tear film, impair ocular surface defenses, and disrupt the retinal nerve fiber layer [10,11]. Similarly, the widespread use of eye cosmetics poses potential risks to ocular health. Ingredients in cosmetics, particularly mascara and internal eyeliner, may alter the lipid layer of the tear film, leading to discomfort and exacerbating DED [12]. While tobacco and cosmetics are predominantly used, limited research has explored their contribution to DED prevalence in the UAE, leaving a critical gap in understanding gender-specific risk factors.

This study seeks to address these gaps by estimating the prevalence of symptomatic DED among university students in the UAE and investigating its association with behavioral and cultural risk factors, such as smoking and cosmetic use, using the validated OSDI questionnaire. Studying the prevalence of symptomatic DED among UAE university students is important for several public health and academic reasons. Younger populations (e.g., university students) are generally healthy, where DED is often underdiagnosed, and may not seek eye care regularly [13]. Additionally, they often engage in lifestyle (e.g., extended screen time, smoking, cosmetic use, contact lens use, and irregular sleep patterns) that increase the risk of DED [14]. Along with the lifestyle factor, UAE has an environmental condition (e.g., hot, dry, and dusty climate) that is known to exacerbate dry eye symptoms [15]. Finally, the prevalence of DED symptoms (e.g., irritation, blurred vision, and eye fatigue) among university students may interfere with academic performance, especially reading, studying, and concentrating during lectures and exams [16]. Understanding the above-mentioned factors and its correlation with DED in UAE students helps tailor prevention and education strategies; prompt early intervention and disease management; and helps estimate how much DED affects student productivity and well-being.

This study is the first to comprehensively examine the prevalence of symptomatic DED and its association with behavioral and cultural risk factors among university students in the United Arab Emirates (UAE). It uniquely investigates the interplay between DED and lifestyle practices, such as smoking behaviors (*e.g.,* Dokha, Ajami), the duration of daily electronic device use, and the number of studying hours per day. Furthermore, the study explores the effect of eye cosmetics use, including long-term application and hygiene practices on DED severity. By focusing on young university students—a demographic increasingly exposed to digital screens, academic stress, and cultural practices, this research fills a critical gap in understanding how these factors influence DED prevalence and severity. Utilizing the English version of the validated OSDI [5] and robust statistical analyses, the study provides novel insights that are expected to inform targeted awareness campaigns and intervention strategies tailored to this population. By providing region-specific data, these research findings will raise awareness among students and inform the development of community-based interventions and awareness campaigns. The findings are expected to guide local authorities, non-governmental organizations, and eye health professionals in designing educational programs to mitigate DED risk factors.

## 2. Methods

### 2.1. Study design and population

A cross-sectional study was conducted between April 2023 and December 2023 to assess DED prevalence among university students in the UAE. Participants were recruited through a Convenience Sampling Technique, with a Google survey distributed via social media platforms and research departments in UAE universities. The study population comprised 654 university students, stratified by variables such as gender, age, residence, field of study, study level, daily study duration, electronic device usage, contact lens wear, use of moisturizing eye drops, and current diagnosis of eye dryness. Inclusion criteria consisted of all university students ≥18 years old studying in the UAE universities. Exclusion criteria included individuals under 18 years old, non-university students, and those with eye surgeries, active ocular diseases, or ophthalmic and systemic medications (antihistamines/decongestants, antidepressants (TCAs, SSRIs), blood pressure meds (beta-blockers, diuretics), hormonal drugs (birth control, HRT), acne treatments (isotretinoin), and sleeping pills, as well as certain pain relievers, Parkinson's drugs, and even some eye drops) affecting tear film production or ocular surface integrity.

### 2.2. Study tool: Ocular surface disease index (OSDI)

The English version OSDI questionnaire, developed by the Outcomes Research Group at Allergan Inc., is a validated instrument designed to assess symptoms related to eye irritation and DED [5]. It consists of 12 items divided into three sections: ocular symptoms, vision-related function, and environmental triggers. Each item is graded on a scale from 0 (none of the time) to 4 (all of the time). The total OSDI score is calculated using the formula:

$$OSDI\ Score = \frac{Sum\ of\ scores\ for\ all\ questions\ answered\ \times\ 100}{Total\ number\ of\ questions\ answered\ \times\ 4}$$

Scores are classified into three categories: normal [0–12], mild-to-moderate [13–32], and severe [33–100].

OSDI scores were summarized using medians and interquartile ranges due to their non-normal distribution, consistent with the use of non-parametric statistical tests.

### 2.3. Assessment of behavioral and cultural risk factors

The study evaluates various behavioral and cultural risk factors associated with the prevalence and severity of DED among the study population. Smoking behaviors assessed include the types of tobacco used—such as regular cigarettes, electronic cigarettes, Medwakh pipes, Mouassal waterpipes (flavored and honeyed tobacco/glycerin paste), and Tumbak/Ajami waterpipes (non-honeyed pure dark tobacco paste)—as well as smoking frequency (daily, weekly/1–2 times per

week, or monthly/1–2 times per month) and smoking environment (indoors or outdoors). Additionally, smoking quantification and duration are examined by considering the number of cigarettes, waterpipes, or Medwakh pipes smoked per week and the total years of smoking. Eye cosmetic use is also analyzed, focusing on the types of products (mascara, eyeliner, eyeshadow, eyelashes), duration of use (<6 months, 6–12 months, >12 months), daily application time (in hours), cleaning methods (water, soapy water, or cleansing products), frequency of night cleansing (never, sometimes, always), and sleep routines (with or without lenses and/or eye cosmetics). These variables are investigated to identify their relationship with the development and severity of DED.

The OSDI questionnaire assessed the severity of DED symptoms, with higher scores indicating more severe symptoms. Behavioral variables included the number of smoking years, frequency and type of eye cosmetics use, and daily habits such as the number of study hours and time spent using electronic devices.

The internal consistency of the behavioral and cultural risk items was assessed using Cronbach's alpha, yielding a value of 0.856. This indicates good internal reliability and supports the use of these as a coherent measure in subsequent analyses.

## 2.4. Data analysis

Data analysis was performed using the Statistical Package for the Social Sciences (SPSS), version 21 (IBM Corp., Chicago, IL, USA). The prevalence of DED was expressed in % of participants with an OSDI score ≥ 13, encompassing mild, moderate, and severe OSDI categories.

The normality of the data was assessed using the Shapiro-Wilk & Kolmogorov-Smirnov tests, and all tested variables were found to be non-normally distributed ($P < 0.05$). Descriptive statistics were calculated using the median and Interquartile Range (IQR) to summarize the non-normally distributed continuous variables. To analyze differences in OSDI scores based on categorical variables (*e.g.,* smoking status and frequency of eye cosmetics use), the Kruskal-Wallis test was applied, with results reported as the median, IQR, H-value, and P-value, where larger H-values indicate greater differences between group medians. Additionally, the Mann-Whitney U test was carried out for the binary variables presented by a U-statistic and Z-score. For both tests, statistical significance is determined by the p-value, with $p < 0.05$ indicating significant differences between groups. The difference between OSDI scores and quantitative variables such as the number of smoking years, study hours, and electronic device usage was assessed using Spearman's rank correlation coefficient, with correlation coefficients ($\rho$) and P-values reported. A significant level of 0.05 ($P < 0.05$) was adopted with 95% confidence intervals reported where applicable.

## 2.5. Ethical considerations

Ethical approval (2023/95) was obtained from the Institutional Review Board (IRB) at Jordan University of Science and Technology (JUST). All participants provided written informed consent to be acknowledged about the benefits, potential risks, and the confidentiality prior to their participation in the study.

## 3. Results

### 3.1. DED and participants' profiles

A total of 654 university students participated in this study. The majority were females (54.1%) and aged between 18 and 21 years (48.6%). Regarding the participants' field and year of study, the highest participation was from undergraduate levels (92.7%) mainly from engineering students (24.6%) and business and tourism students (29.1%). Most participants were non-smokers (58.9%) and did not wear contact lenses (66.2%). 82% of the participants had not been clinically diagnosed with DED. According to the symptomatic assessment of DED by OSDI, 47.1% of the participants (47.1%) reported mild-to-severe DED (Table 1).

**Table 1. Kruskal-Wallis test results for the demographics section.**

| Variable | | Frequency (%) | Median OSDI (IQR) | Kruskal-Wallis Test* |
|---|---|---|---|---|
| **Age Interval** | **18-20** | 318 (48.6%) | 12.50 (18.75) | H = 2.026 P = 0.567 |
| | **21-23** | 243 (37.2%) | 12.50 (18.75) | |
| | **24-26** | 63 (9.6%) | 14.58 (22.95) | |
| | **≥ 27** | 30 (4.6%) | 13.54 (31.77) | |
| **Field of Study** | **Natural Sciences** | 19 (2.9%) | 12.50 (18.75) | H = 13.622 P = 0.058 |
| | **Health Sciences** | 83 (12.7%) | 12.50 (22.92) | |
| | **Medical and Pharmaceutical Sciences** | 28 (4.3%) | 13.54 (30.20) | |
| | **Engineering Sciences** | 161 (24.6%) | 10.42 (16.66) | |
| | **Art and Media Sciences** | 127 (19.4%) | 16.67 (22.92) | |
| | **Business and Tourism** | 190 (29.1%) | 12.50 (20.83) | |
| | **Education and Social Sciences** | 38 (5.8%) | 15.62 (15.63) | |
| | **Humanities and Political Sciences** | 8 (1.2%) | 5.21 (9.37) | |
| **Level of Study** | **Bachelor (1ˢᵗ year)** | 184 (28.1%) | 11.45 (16.67) | H = 6.649 P = 0.355 |
| | **Bachelor (2ⁿᵈ year)** | 137 (20.9%) | 12.50 (22.91) | |
| | **Bachelor (3ʳᵈ year)** | 171 (26.1%) | 12.50 (18.75) | |
| | **Bachelor (4ᵗʰ year)** | 115 (17.6%) | 14.58 (27.08) | |
| | **Master (1ˢᵗ year) or equivalent** | 43 (6.6%) | 12.50 (22.91) | |
| | **Master (2ⁿᵈ year) or equivalent** | 3 (0.5%) | 12.50 | |
| | **Ph.D. or equivalent** | 1 (0.2%) | --- | |
| **Contact Lens (CL) use** | **Never** | 433 (66.2%) | 10.42 (16.66) | H = 39.986 **P < 0.001** |
| | **Sometimes** | 157 (24.0%) | 18.75 (22.92) | |
| | **Always** | 64 (9.8%) | 22.92 (27.08) | |
| **Eye Dryness** | **No** | 536 (82.0%) | 10.42 (18.75) | H = 56.664 **P < 0.001** |
| | **Yes** | 118 (18.0%) | 25.00 (31.25) | |
| **OSDI Status** | **Normal** | 346 (52.9%) | | |
| | **Mild-to-Moderate** | 193 (29.5%) | | |
| | **Severe** | 115 (17.6%) | | |

| Variable | | Frequency (%) | Median OSDI (IQR) | Mann-Whitney U |
|---|---|---|---|---|
| **Gender** | **Male** | 300 (45.9%) | 10.42 (16.66) | U = 44,579.5 Z = −3.545 **P < 0.001** |
| | **Female** | 354 (54.1%) | 14.58 (22.92) | |
| **Smoking Habits** | **Non-smokers** | 385 (58.9%) | 10.42 (18.75) | H = 40,348.0 Z = −4.818 **P < 0.001** |
| | **Smokers** | 269 (41.1%) | 16.67 (25.00) | |

\* Values in bold indicate significant difference (P < 0.05).

To analyze differences in OSDI scores across categorical variables, such as gender, age group, field of study, and level of education, the Kruskal-Wallis test was employed, due to non-normal data distribution, to examine the if significant differences existed in OSDI scores among variables. A Mann-Whiteny U test was performed for the binary variables (gender and smoking status). The results are presented in Table 1.

The Mann–Whitney U test demonstrated a statistically significant difference in OSDI scores between male and female participants (U = 44,579.50, Z = −3.545, P < 0.001). Female participants exhibited higher OSDI scores than males, as reflected by higher mean ranks (351.57 vs. 299.10), indicating greater severity of dry eye symptoms among females.

Similarly, OSDI scores differed significantly according to smoking status. Smokers reported significantly higher OSDI scores compared to non-smokers (U=40,348.00, Z= −4.818, P <0.001). The mean rank of OSDI scores was higher among smokers (370.01) than non-smokers (297.80), suggesting that tobacco use is associated with increased ocular surface discomfort.

The Kruskal-Wallis test showed a significant gender difference in OSDI scores (H=12.569, P<0.001), with females reporting higher median scores (14.58, IQR: 22.92) than males (10.42, IQR: 16.66). In contrast, no significant differences were found across age groups (H=2.026, P=0.567), with median OSDI scores ranging from 12.50 (IQR: 18.75) in the 18–23 group to 14.58 (IQR: 22.95) in the 24–26 group, and 13.54 (IQR: 31.77) among participants aged ≥ 27.

The field of study showed no significant differences in OSDI scores (H=13.622, P=0.058), although Art and Media Sciences had the highest median score (16.67, IQR: 22.92) and Humanities and Political Sciences the lowest (5.21, IQR: 9.37). Likewise, the level of study showed no significant variation (H=6.649, P=0.355), with median OSDI scores ranging from 11.45 (IQR: 16.67) in first-year students to 14.58 (IQR: 27.08) in fourth-year students.

A significant difference existed in OSDI scores among smoking habits (H=26.721, P<0.001), as smokers had higher median scores (16.67, IQR: 25.00) compared to non-smokers (10.42, IQR: 18.75). Contact lens use was also significantly different (H=39.986, P<0.001), with those always using lenses reporting the highest median OSDI score (22.92, IQR: 27.08), compared to those who sometimes (18.75, IQR: 22.92) or never (10.42, IQR: 16.66) used lenses.

When the Kruskal-Wallis test indicated a statistically significant difference among variables with more than two categories, post hoc pairwise comparisons were conducted using Dunn's test with Bonferroni adjustment to control for multiple testing. Finally, eye dryness was significantly related to OSDI scores (H=56.664, P<0.001). Participants reporting eye dryness had markedly higher median scores (25.00, IQR: 31.25) than those without dryness (10.42, IQR: 18.75).

Post hoc pairwise comparisons using Dunn's test with Bonferroni adjustment revealed that participants who never used contact lenses had significantly lower OSDI scores compared to both those who sometimes used lenses (adjusted P<0.001) and those who always used lenses (adjusted p<0.001). The difference between sometimes and always users was not statistically significant after Bonferroni correction (adjusted P=0.469).

### 3.2. Smoking profile

The Kruskal-Wallis test was conducted to assess if there is a significant difference in OSDI scores among smoking variables. This analysis included all 654 study participants, comprising 269 smokers (41.1%) and 385 non-smokers (58.9%), as detailed in Table 1. For each smoking-related variable, participants were categorized based on their reported smoking behaviors, and OSDI scores were compared across these categories. The results are presented in Table 2.

There was a significant difference in OSDI scores among smoking types. (H=31.688, P<0.001). Non-smokers reported the lowest median OSDI score (4.17, IQR: 10.42), while smokers using Dokha (20.83, IQR: 31.25) and waterpipe (Mouassal) (20.83, IQR: 39.59) had the highest scores. Regular cigarette smokers had a median score of 16.67 (IQR: 35.41), electronic cigarette (Vape) users had 14.58 (IQR: 18.75), and waterpipe (Ajami) users reported 10.42 (IQR: 29.16).

The smoking rate also significantly influenced OSDI scores (H=25.737, P<0.001). Participants who smoked daily had the highest median OSDI score (17.71, IQR: 29.69), followed by weekly smokers (16.67, IQR: 21.36) and monthly smokers (12.50, IQR: 14.59), with non-smokers reporting a median score of 10.42 (IQR: 18.75).

Smoking areas were significantly associated with OSDI scores (H=34.491, P<0.001). Participants who smoked indoors reported the highest median OSDI score (27.08, IQR: 31.76), followed by those who smoked in mixed areas (16.17, IQR: 22.40) and outdoor smokers (12.50, IQR: 20.83). Non-smokers again had the lowest scores, with a median of 10.42 (IQR: 18.75).

### 3.3. The usage of eye cosmetic products

The Kruskal-Wallis test was performed to evaluate if significant differences exist in OSDI scores among eye cosmetics usages. This analysis included all 654 study participants. Results are presented in Table 3.

**Table 2. Kruskal-Wallis test results for the smoking profile.**

| Variable | | Median OSDI (IQR) | Kruskal-Wallis Test* |
|---|---|---|---|
| Smoking Types | Regular Cigarette | 16.67 (35.41) | H = 31.688 **P < 0.001** |
| | Electronic Cigarette (vape) | 14.58 (18.75) | |
| | Dokha (medwakh pipe) | 20.83 (31.25) | |
| | Waterpipe (mouassal) | 20.83 (39.59) | |
| | Waterpipe (ajami) | 10.42 (29.16) | |
| | Non-smoker | 4.17 (10.42) | |
| Smoking Rate | Daily | 17.71 (29.69) | H = 25.737 **P < 0.001** |
| | Weekly (1–2 times per week) | 16.67 (21.36) | |
| | Monthly (1–2 times per month) | 12.50 (14.59) | |
| | Non-smoker | 10.42 (18.75) | |
| Smoking Areas | Indoor | 27.08 (31.76) | H = 34.491 **P < 0.001** |
| | Outdoor | 12.50 (20.83) | |
| | Mixed | 16.17 (22.40) | |
| | Non-Smoker | 10.42 (18.75) | |

* Values in bold indicate significant difference (P < 0.05).

Inner eyeliner use showed a **significant effect** on OSDI scores (H = 17.329, P = 0.002). Daily users had the highest median score (19.79, IQR: 27.09), followed by those applying it 1–2 times per week (16.67, IQR: 28.64), 1–2 times per month (16.67, IQR: 21.87), and 3–4 times per week (14.58, IQR: 15.62). Participants who never used inner eyeliner reported the lowest median score (12.50, IQR: 18.75).

External eyeliner usage also showed a significant effect on OSDI scores (H = 23.263, P < 0.001). Daily users had a median score of 16.67 (IQR: 26.04), similar to those using it 3–4 times per week (16.67, IQR: 28.12) or 1–2 time(s) per week (15.63, IQR: 29.17). Occasional users (1–2 time(s) per month) reported a median of 12.50 (IQR: 20.83), while those who never used external eyeliner had the lowest scores (10.42, IQR: 16.66).

False eyelash usage was significantly associated with OSDI scores (H = 22.444, P < 0.001). Participants using false eyelashes 3–4 times per week had the highest median score (33.33, IQR: 40.63), followed by those using them 1–2 time(s) per month (21.87, IQR: 28.13) or 1–2 time(s) per week (20.83, IQR: 29.17). Daily users reported a median of 14.58 (IQR: 48.96), and non-users had the lowest scores (12.50, IQR: 16.67).

Eye shadow usage was another significant factor (H = 17.001, P = 0.002). Daily users had the highest median OSDI score (17.71, IQR: 24.48), followed by 1–2 time(s) per week users (16.67, IQR: 26.04), and 1–2 time(s) per month users (14.58, IQR: 20.32). Participants using eye shadow 3–4 times per week had a median score of 12.50 (IQR: 22.92), while non-users reported the lowest scores (10.42, IQR: 17.18).

Significant differences also existed in OSDI scores among Mascara usage (H = 20.234, P < 0.001). Participants using mascara 1–2 time(s) per month had the highest median score (18.75, IQR: 37.50), followed by 1–2 time(s) per week users (16.67, IQR: 31.25), daily users (14.58, IQR: 22.92), and 3–4 times per week users (14.58, IQR: 18.75). Non-users had the lowest scores (10.42, IQR: 16.66).

### 3.4. Eye cosmetics routines and practices

The Kruskal-Wallis test was conducted to examine the difference between eye cosmetics practices and OSDI scores. This analysis included all 654 study participants, with practice variables assessed among those who reported using eye cosmetics. Given the observed gender differences in OSDI scores and the gender-specific nature of eye cosmetic use,

**Table 3. Kruskal-Wallis test results for the usage of the eye cosmetics products.**

| Variable | | Frequency (%) | Median OSDI (IQR) | Kruskal-Wallis Test* |
|---|---|---|---|---|
| **Inner Eyeliner** | **Daily** | 36 (5.5) | 19.79 (27.09) | H = 17.329 **P = 0.002** |
| | **3-4 times per week** | 30 (4.6) | 14.58 (15.62) | |
| | **1-2 times per week** | 68 (10.4) | 16.67 (28.64) | |
| | **1-2 times per month** | 77 (11.8) | 16.67 (21.87) | |
| | **Never/NA** | 443 (67.7) | 12.50 (18.75) | |
| **External Eyeliner** | **Daily** | 61 (9.3) | 16.67 (26.04) | H = 23.263 **P < 0.001** |
| | **3-4 times per week** | 54 (8.3) | 16.67 (28.12) | |
| | **1-2 times per week** | 68 (10.4) | 15.63 (29.17) | |
| | **1-2 times per month** | 75 (11.5) | 12.50 (20.83) | |
| | **Never/NA** | 396 (60.6) | 10.42 (16.66) | |
| **False Eyelashes** | **Daily** | 14 (2.1) | 14.58 (48.96) | H = 22.444 **P < 0.001** |
| | **3-4 times per week** | 14 (2.1) | 33.33 (40.63) | |
| | **1-2 times per week** | 20 (3.1) | 20.83 (29.17) | |
| | **1-2 times per month** | 54 (8.3) | 21.87 (28.13) | |
| | **Never/NA** | 552 (84.4) | 12.50 (16.67) | |
| **Eye shadow** | **Daily** | 44 (6.7) | 17.71 (24.48) | H = 17.001 **P = 0.002** |
| | **3-4 times per week** | 43 (6.6) | 12.50 (22.92) | |
| | **1-2 times per week** | 77 (11.8) | 16.67 (26.04) | |
| | **1-2 times per month** | 100 (15.3) | 14.58 (20.32) | |
| | **Never/NA** | 390 (59.6) | 10.42 (17.18) | |
| **Mascara** | **Daily** | 126 (19.3) | 14.58 (22.92) | H = 20.234 **P < 0.001** |
| | **3-4 times per week** | 90 (13.8) | 14.58 (18.75) | |
| | **1-2 times per week** | 63 (9.6) | 16.67 (31.25) | |
| | **1-2 times per month** | 39 (6.0) | 18.75 (37.50) | |
| | **Never/NA** | 336 (51.4) | 10.42 (16.66) | |

* Values in bold indicate significant difference (P < 0.05).

additional gender-stratified analyses were conducted. Eye cosmetic practice variables were primarily evaluated among female participants, in whom cosmetic use was prevalent, while male participants were retained in the analysis as a reference group. The stratified analyses confirmed that the associations between eye cosmetic practices and OSDI scores were driven predominantly by female participants. Results are presented in Table 4.

The duration of cosmetics use per year significantly influenced OSDI scores (H = 22.628, P < 0.001). Participants using cosmetics for 6–12 months reported the highest median score (18.75, IQR: 9.17), followed by those using cosmetics for less than 6 months (16.67, IQR: 25.00) or more than 12 months (14.58, IQR: 20.83). Those who did not use cosmetics or had no applicable data had the lowest scores (10.42, IQR: 16.66).

The duration of cosmetics use per day also showed a significant effect (H = 18.821, P < 0.001). Participants using cosmetics for less than 6 hours daily had a median score of 16.67 (IQR: 20.32), while those using cosmetics for 6–12 hours or more than 12 hours had scores of 14.58 (IQR: 33.33) and 14.58 (IQR: 20.83), respectively. Non-users or those without applicable data had the lowest scores (10.42, IQR: 16.66).

Cleansing techniques were significantly associated with OSDI scores (H = 22.990, P < 0.001). Participants using only water reported the highest median score (17.71, IQR: 34.90), followed by those using cleansing cream (14.58, IQR: 22.92) and soap and water (12.50, IQR: 18.75). Non-users or those without applicable data had the lowest scores (10.42, IQR: 16.66).

**Table 4. Kruskal-Wallis test results for eye cosmetics routines and practices.**

| Variable | | Frequency (%) | Median OSDI (IQR) | Kruskal-Wallis Test* |
|---|---|---|---|---|
| Duration of Cosmetics use (per year) | < 6 months | 51 (7.8) | 16.67 (25.00) | H = 22.628 **P < 0.001** |
| | 6-12 months | 53 (7.8) | 18.75 (9.17) | |
| | >12 months | 222 (33.9) | 14.58 (20.83) | |
| | Don't use or NA | 328 (50.2) | 10.42 (16.66) | |
| Duration of Cosmetics use (per day) | < 6 hours | 100 (15.3) | 16.67 (20.32) | H = 18.821 **P < 0.001** |
| | 6-12 hours | 110 (16.8) | 14.58 (33.33) | |
| | >12 hours | 88 (13.5) | 14.58 (20.83) | |
| | Don't use or NA | 356 (54.4) | 10.42 (16.66) | |
| Cleansing Technique | Cleansing Cream | 260 (39.8) | 14.58 (22.92) | H = 22.990 **P < 0.001** |
| | Soap Water | 47 (7.2) | 12.50 (18.75) | |
| | Water ONLY | 36 (5.5) | 17.71 (34.90) | |
| | Don't use or NA | 311 (47.6) | 10.42 (16.66) | |
| Cleansing Rate (per day) | Rarely | 257 (39.3) | 14.58 (20.83) | H = 35.622 **P < 0.001** |
| | Sometimes | 51 (7.8) | 14.58 (31.25) | |
| | Always | 25 (3.8) | 33.33 (27.08) | |
| | Don't use or NA | 321 (49.1) | 10.42 (16.66) | |
| Sleep Routine | Sleep with lenses | 4 (0.6) | 28.13 (33.85) | H = 6.937 P = 0.074 |
| | Sleep with eye cosmetics | 6 (0.9) | 23.96 (40.62) | |
| | Sleep with eye cosmetics and lenses | 5 (0.8) | 52.08 (67.71) | |
| | Sleep with no lenses and eye cosmetics or NA | 639 (97.7) | 12.50 (18.75) | |

*Values in bold indicate significant difference (P < 0.05).

The rate of cleansing per day was another significant factor (H = 35.622, P < 0.001). Participants who always cleansed their cosmetics reported the highest median OSDI score (33.33, IQR: 27.08), followed by those cleansing sometimes (14.58, IQR: 31.25) or rarely (14.58, IQR: 20.83). Non-users or those with no applicable data had the lowest scores (10.42, IQR: 16.66).

Regarding sleep routines, although not statistically significant (H = 6.937, P = 0.074), participants who slept with both eye cosmetics and lenses had the highest OSDI scores (52.08, IQR: 67.71), followed by those who slept only with eye cosmetics (23.96, IQR: 40.62) and those who slept with lenses (28.13, IQR: 33.85). Participants who slept without lenses and cosmetics or had no applicable data had the lowest median scores (12.50, IQR: 18.75).

### 3.5. Statistical correlation among OSDI score, smoking, studying, and electronic use

This analysis included all 654 study participants, though the actual number of cases varied for specific variables based on participant responses. For instance, smoking-related variables (number of smoking years, cigarettes per week, Dokha per week, Shisha per week) were assessed only among smokers (n = 269), while electronic device usage and study hours were assessed among all participants (n = 654).

The results of Spearman's rank correlation analysis indicate several significant associations between the OSDI score, and the different variables as presented in Table 5.

There is a strong positive correlation between OSDI scores and daily electronic device use (r = 0.694, P < 0.001), indicating that longer screen time is associated with greater eye dryness and discomfort. A similarly strong positive correlation is observed between OSDI scores and hours spent studying per day (r = 0.661, P < 0.001), suggesting that extended study time is also linked to increased eye discomfort.

**Table 5. Spearman's rank correlation results.**

| Dependent Variable | Independent Variables | Correlation Coefficient | P-value* |
|---|---|---|---|
| OSDI Score | Electronic device usage per day (in hours) | 0.694 | **<0.001** |
| | Studying hours per day | 0.661 | **<0.001** |
| | Number of smoking years | 0.380 | **<0.001** |
| | Number of cigarettes smoked per week | 0.175 | **<0.001** |
| | Number of Dokha smoked per week | 0.005 | 0.896 |
| | Number of Shisha smoked per week | 0.097 | **0.013** |

*Values in bold indicate significant difference (P<0.05).

The analysis shows a moderate positive correlation between OSDI scores and the number of years a person has been smoking (r=0.38, P<0.001), indicating that longer smoking history is associated with greater eye discomfort. In comparison, the correlation between OSDI scores and the number of cigarettes smoked per week is weak but statistically significant (r=0.175, P<0.001), suggesting only a slight increase in eye discomfort with higher cigarette consumption.

The relationship between OSDI scores and Dokha use is negligible and not significant (r=0.005, P=0.896), indicating no meaningful effect. For Shisha smoking, the correlation is weak yet significant (r=0.097, P=0.013), implying a small but measurable association with increased eye discomfort.

Overall, the strongest predictors of higher OSDI scores remain electronic device usage and studying hours, both showing strong positive correlations. Smoking contributes to eye discomfort primarily through years of smoking, while Dokha shows no effect and Shisha shows only a weak association.

## 4. Discussion

The discussion of these findings can delve into several key aspects regarding the association between smoking behaviors and OSDI scores. The significant associations between smoking profiles and OSDI scores highlight the potential impact of various smoking behaviors on dry eye symptoms and severity.

### 4.1. Gender, smoking habits, and eye dryness

The results reveal a gender difference, with females reporting higher OSDI scores than males. This may be attributed to hormonal differences, which are known to influence tear film stability and the prevalence of DED [17,18]. Smoking habits also demonstrated a strong association with OSDI scores, where smokers had significantly higher scores than non-smokers [19]. Among smoking types, Dokha and Mouassal waterpipe users exhibited the highest median OSDI scores. This aligns with existing literature that links smoking to oxidative stress, reduced tear film quality, and inflammation, exacerbating dry eye symptoms [20,21]. Both Dokha and Mouassal (waterpipe/shisha) are associated with high oxidative stress, and in some contexts, they may produce equal or greater oxidative stress than cigarettes, though the mechanisms and exposure patterns differ. Dokha is often more intense per session than cigarette smoking due to its very high nicotine content (strong stimulation of oxidative pathways); rapid and deep inhalation (sharp spikes in reactive oxygen species; and its positive correlation with an oxidative stress marker (e.g., lipid peroxidation) [22]. Additionally, Mouassal smoking causes prolonged and cumulative oxidative stress along with high exposure to carbon monoxide, polycyclic aromatic hydrocar (PAHs), and heavy metals that would generate systemic oxidative stress and inflammation [23].

### 4.2. Frequency and environment of smoking

Daily smokers reported the highest OSDI scores compared to weekly (1–2 times per week) or monthly (1–2 times per month) smokers, indicating a dose-response relationship between smoking frequency and the severity of ocular

discomfort [24]. Moreover, smoking in indoor environments led to the highest OSDI scores. This underscores the compounded effect of poor ventilation and higher exposure to smoke particulates in closed settings, which could aggravate ocular surface damage [25].

The results emphasize the importance of smoking cessation and mitigating indoor smoking environments as preventive measures for reducing DED risk. Public health interventions could target high-risk groups, such as frequent smokers and individuals exposed to smoke indoors. Furthermore, raising awareness about the detrimental ocular effects of smoking, including specific tobacco products like Dokha and Mouassal, could aid in behavioral changes.

### 4.3. Eye cosmetics routine and practices

Research indicates that the use of eye cosmetics can influence ocular comfort and may be associated with dry eye symptoms. A study by Ng et al. (2012) found that while OSDI scores were similar between cosmetics users and non-users, perceived comfort was greater when cosmetics were not used [26]. Users with shorter annual usage durations (< 6 months) reported higher OSDI scores compared to those using cosmetics for over a year. This could be attributed to long-term users' adaptation or improved selection of products compatible with sensitive eyes. Non-users consistently showed the lowest OSDI scores, reinforcing the idea that avoiding cosmetics altogether reduces the risk of ocular surface discomfort.

Surprisingly, extended daily use (> 12 hours) did not significantly elevate OSDI scores compared to moderate use (6–12 hours). This might suggest that discomfort is more influenced by the quality of products and application techniques than by duration alone. Non-users reported the lowest discomfort, consistent with the protective effect of avoiding cosmetics.

The cleansing method significantly influenced OSDI scores. Participants who used water alone for removal experienced the highest discomfort, likely due to the inadequate removal of cosmetic residues, leading to prolonged irritation [12,27]. Conversely, those using soap and water or cleansing creams reported lower scores, suggesting that effective cleansing minimizes residue buildup and irritation.

Frequent cleansing (*e.g.,* "always") was associated with the highest OSDI scores, potentially due to overexposure to harsh chemicals or mechanical irritation from repeated rubbing. This finding highlights the need for balanced hygiene practices that avoid excessive cleansing while ensuring proper removal of cosmetics [3].

Although not statistically significant, the trend indicates that sleeping with both cosmetics and lenses led to the highest discomfort, underscoring the importance of proper removal before rest. This behavior likely exacerbates irritation by trapping residues and reducing tear film replenishment during sleep.

### 4.4. Lifestyle factors

A strong positive correlation between screen time and OSDI scores reflects the well-documented impact of digital eye strain. Prolonged exposure to screens reduces blink rates and destabilizes the tear film, leading to dryness and discomfort [28].

Similar to device usage, extended study hours were strongly correlated with higher OSDI scores. This may be due to sustained focus on reading or screens, further exacerbating dry eye symptoms.

Smoking, particularly over a longer duration, was moderately associated with higher OSDI scores. Cigarette smoke contains irritants that can destabilize the tear film and increase oxidative stress, both of which contribute to ocular surface discomfort. Shisha smoking showed a weaker correlation, possibly due to less frequent usage, while Dokha smoking had no significant impact, perhaps due to differences in exposure levels or study sample characteristics.

This study has several limitations. Data collection was based on an online self-reported questionnaire, which may introduce response bias or social desirability effects, potentially influencing the findings. Also, The estimation of DED was entirely subjective and may not align with clinical sign. Therefore, further research is recommended to clinically assess the prevalence and severity of DED and explore its statistical association with tobacco and cosmetic use.

## 5. Conclusion

This study provides valuable insights into the associations between eye cosmetics practices, smoking behaviors, lifestyle factors, and ocular surface discomfort, as reflected by OSDI scores. Key findings include the role of gender, smoking frequency, and environmental factors in exacerbating symptoms of DED. Additionally, improper cleansing routines and prolonged exposure to screens further contributed to higher OSDI scores. These results underscore the multifactorial nature of ocular surface discomfort and emphasize the importance of public awareness and preventive measures.

Recommendations for future interventions include promoting smoking cessation, advocating for improved indoor air quality, and encouraging balanced hygiene practices for cosmetics users. Public health campaigns targeting high-risk groups, such as smokers and individuals with extended screen time or poor eye care habits, could help mitigate the prevalence and severity of DED.

## Supporting information

**S1 File. Study tool (questionnaire).**
(DOCX)

## Acknowledgments

The authors gratefully acknowledge Prof. May Bakkar for her technical support for the work. Acknowledgment is also given to the Canadian University Dubai (CUD) for approving and supervising the study design and protocol. Sincere gratitude is extended to all participants who contributed to this study.

## Author contributions

**Conceptualization:** Mona Aridi, Nisreen Alwan, Wissam Ghach.

**Data curation:** Mona Aridi, Wissam Ghach.

**Formal analysis:** Mona Aridi, Nisreen Alwan, Wissam Ghach.

**Investigation:** Nisreen Alwan.

**Methodology:** Mona Aridi, Nisreen Alwan, Wissam Ghach.

**Project administration:** Wissam Ghach.

**Supervision:** Wissam Ghach.

**Writing – original draft:** Mona Aridi.

**Writing – review & editing:** Nisreen Alwan, Wissam Ghach.

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
