## [Decision Letter · Decision Letter 0]

8 Dec 2025

Dear Dr. Ghach,

Thank you for submitting your manuscript to PLOS ONE. After careful consideration, we feel that it has merit but does not fully meet PLOS ONE’s publication criteria as it currently stands. Therefore, we invite you to submit a revised version of the manuscript that addresses the points raised during the review process.

We look forward to receiving your revised manuscript.

Kind regards,

Yalong Dang

Academic Editor

PLOS One

Journal Requirements:

3. Please include captions for your Supporting Information files at the end of your manuscript, and update any in-text citations to match accordingly. Please see our Supporting Information guidelines for more information: http://journals.plos.org/plosone/s/supporting-information .

Reviewer's Responses to Questions

**Comments to the Author**

1. Is the manuscript technically sound, and do the data support the conclusions?

Reviewer #1: Partly

Reviewer #2: Partly

Reviewer #3: Yes

Reviewer #4: Yes

2. Has the statistical analysis been performed appropriately and rigorously?

Reviewer #1: Yes

Reviewer #2: No

Reviewer #3: Yes

Reviewer #4: Yes

3. Have the authors made all data underlying the findings in their manuscript fully available?

Reviewer #1: Yes

Reviewer #2: Yes

Reviewer #3: Yes

Reviewer #4: Yes

4. Is the manuscript presented in an intelligible fashion and written in standard English?

Reviewer #1: Yes

Reviewer #2: Yes

Reviewer #3: Yes

Reviewer #4: Yes

Reviewer #1: The research itself is well conducted and analyzed. Although the most common causes of dry eye have been addressed in it, there are several factors that are not considered. For instance, due to the particular geographic area, the time frame could have been wider so it could include cooler months where air conditioning and tear evaporation associated with heat could change the symptoms experienced. In the exclusion criteria it mentions dry eye disease currently under treatment and medications that may cause dry eye. It would be useful to specify if such medications are only ophthalmic or also systemic for other conditions such as the use of isotretinoin for acne, or the use of diphenhydramine for allergic disease. Other than that, I have no further observations. Best regards.

Reviewer #2: This study is interesting and possesses the potential for international publication. However, it requires revisions to significantly improve its clarity

1. Why study students in the UAE? The statement in lines 70–72 refers to a limitation: that studies in the UAE are insufficient for international publication, so the author should add more detail. I agree that this provides region-specific data, but why do we need it from your country?

2. In line 88, the statement refers to the use of a validated OSDI. Please provide the corresponding citation for this instrument. Could you please specify which language version of the OSDI questionnaire was used?

3. Is the author confident that the total number of students in the entire university is only 654, as indicated in line 96?

4. Please add the inclusion criteria in 2.1 subject design and population

5. Have you assessed the internal reliability (or consistency) of the questionnaire items regarding behavioral and cultural risk factors? If yes, please provide the results in Section 2.3.

6. Line 153 indicates that 654 students participated. Could the authors confirm if this figure represents the entire target population, or if it is the number of participants who completed the questionnaire? Clarification is needed on whether this constitutes a 100% response rate.

7. In the Data Analysis section, the Kruskal-Wallis test is a multiple comparison test. Therefore, the specific post hoc analysis method used (e.g., Dunn's test, Conover's test) must be specified.

8. In the Results section, the Kruskal-Wallis test is statistically inappropriate for identifying the association between variables. The fundamental hypothesis of this test is to determine if there are significant differences between the medians of three or more independent groups, not to measure the strength or direction of an association. Therefore, the authors should apply an appropriate test of association. (i.e. line 201, 236)

9. The presentation of results is overly lengthy and verbose. I recommend condensing this section for better clarity and impact

10. The results section is unclear. Specifically, please explain the methodology used to analyze the data presented in Tables 3.2 and 3.3. Furthermore, please specify the exact number of participants included in the analysis for these tables.

11. Contact lens wear is an established associated factor for dry eye and therefore constitutes a potential confounder in this study. Could the authors explain their rationale for not excluding contact lens wearers from the study?

12. I question whether the analysis for this specific area should be restricted to female participants only. If male participants were included, this must be clearly justified and described in the Methodology (Data Analysis section). Furthermore, since Table 1 indicates a difference in OSDI scores between male and female participants, a stratified analysis by gender should be considered.

13. In Table 1, the comparison of OSDI scores between genders should employ a different statistical test

Reviewer #3: This is very good study to carry on. It would have been better to include clinical measurements as well. OSDI score is very important tool for evaluating dry Eye diseases but without clinical correlation, it is still not significant.

Reviewer #4: First of all, your research had very detailed information and I enjoyed reading your manuscript. Thank you for describing different lifestyle and cultures in the Middle East for the reader to have better understanding of your research purposes and outcomes. However, I have the following comments and questions:

1. Regarding the cosmetic use, you have analysed each type of cosmetic as an isolated factor. However, I believe that many individuals have worn more than one type of cosmetic on their eyes, including eyeliner and mascara. Do you think there is an alternative analysis for those associated/multivariable factors, rather than the analysis of each factor separately? Also, the preference for cosmetic use in females, but not males, may encourage the subgroup analysis of female participants in this issue.

2. Due to the design of the questionnaire survey, please also describe the response rate of the questionnaire in each field, or describe how the questionnaire is designed to fill all questions as a mandatory rule to submit. Additionally, please also discuss the pros and cons of using a questionnaire.

3. Moreover, you have described/quantified many habits as longer, more frequent, shorter. Could you please describe them more in quantifiable measures? Also does weekly and monthly smokers mean once a week and once a month, respectively?

4. In lines 301 to 303, you emphasize the findings of “Dokha and Mouassal waterpipe users exhibited the highest median OSDI scores” and continuously describe the previous findings on smoking and oxidative stress. Does this specific type of smoking cause more oxidative stress than others?

5. The cleansing methods also prompt a question to me when you describe “Frequent cleansing or always” as a subgroup. Is it a usual habit to leave the cosmetics uncleaned for longer than a day? This is quite interesting in terms of cultural differences, as you mentioned.

6. I believe that many readers are unfamiliar with the H-value, please could you explain more in the manuscript on how to interpret the findings, in association with P-value, so all readers can follow your manuscript with more understanding?

**Do you want your identity to be public for this peer review?** For information about this choice, including consent withdrawal, please see our Privacy Policy

Reviewer #1: No

Reviewer #2: **Yes:** Teera Poyomtip

Reviewer #3: **Yes:** Raju Kaiti

Reviewer #4: No

---

## [Author Response · Author response to Decision Letter 1]

29 Dec 2025

Reviewer #1: The research itself is well conducted and analyzed. Although the most common causes of dry eye have been addressed in it, there are several factors that are not considered. For instance, due to the particular geographic area, the time frame could have been wider so it could include cooler months where air conditioning and tear evaporation associated with heat could change the symptoms experienced. In the exclusion criteria it mentions dry eye disease currently under treatment and medications that may cause dry eye. It would be useful to specify if such medications are only ophthalmic or also systemic for other conditions such as the use of isotretinoin for acne, or the use of diphenhydramine for allergic disease. Other than that, I have no further observations. Best regards.

Answer: Thank you for your valuable feedback. Authors agree with reviewer 1 “Although the most common causes of dry eye have been addressed in it, there are several factors that are not considered. For instance, due to the particular geographic area, the time frame could have been wider so it could include cooler months where air conditioning and tear evaporation associated with heat could change the symptoms experienced”. Kindly note that data collection occurred between April and Dec where both very hot weather (Apil-October) and cooler weather (Nov-Dec) were considered to include different climate conditions. Nevertheless, the climate factor was not considered in our statistics. In the future studies, climate factor will be considered as recommended.

Revised paragraph “Exclusion criteria included individuals under 18 years old, non-university students, and those with eye surgeries, active ocular diseases, or ophthalmic and systemic medications (antihistamines/decongestants, antidepressants (TCAs, SSRIs), blood pressure meds (beta-blockers, diuretics), hormonal drugs (birth control, HRT), acne treatments (isotretinoin), and sleeping pills, as well as certain pain relievers, Parkinson's drugs, and even some eye drops) affecting tear film production or ocular surface integrity”.

Reviewer #2: This study is interesting and possesses the potential for international publication. However, it requires revisions to significantly improve its clarity

1. Why study students in the UAE? The statement in lines 70–72 refers to a limitation: that studies in the UAE are insufficient for international publication, so the author should add more detail. I agree that this provides region-specific data, but why do we need it from your country?

Answer: “This study seeks to address these gaps by estimating the prevalence of symptomatic DED among university students in the UAE and investigating its association with behavioral and cultural risk factors, such as smoking and cosmetic use, using the validated OSDI questionnaire. Studying the prevalence of symptomatic DED among UAE university students is important for several public health and academic reasons. Younger populations (e.g., university students) are generally healthy, where DED is often underdiagnosed, and may not seek eye care regularly (ref 13). Additionally, they often engage in lifestyle (e.g., extended screen time, smoking, cosmetic use, contact lens use, and irregular sleep patterns) that increase the risk of DED (ref 14). Along with the lifestyle factor, UAE has an environmental condition (e.g., hot, dry, and dusty climate) that is known to exacerbate dry eye symptoms (ref 15). Finally, the prevalence of DED symptoms (e.g., irritation, blurred vision, and eye fatigue) among university students may interfere with academic performance, especially reading, studying, and concentrating during lectures and exams (ref 16). Understanding the above-mentioned factors and its correlation with DED in UAE students helps tailor prevention and education strategies; prompt early intervention and disease management; and helps estimate how much DED affects student productivity and well-being”.

2. In line 88, the statement refers to the use of a validated OSDI. Please provide the corresponding citation for this instrument. Could you please specify which language version of the OSDI questionnaire was used?

Answer: Revised “Utilizing the English version of the validated OSDI (ref 5) and robust statistical analyses, the study provides novel insights that are expected to inform targeted awareness campaigns and intervention strategies tailored to this population”.

3. Is the author confident that the total number of students in the entire university is only 654, as indicated in line 96?

Answer: Depending on the utilized strategy of data collection, authors confirms that all participants were university students recruited throughout social medial platforms and research departments of the local universities in the UAE. Table 1 data also confirms that most of the participants are either undergraduate or master 1 students. The number does not refer to total number of students in the University.

4. Please add the inclusion criteria in 2.1 subject design and population

Answer: Added “Inclusion criteria: university students ≥18 years old studying in the UAE universities”.

5. Have you assessed the internal reliability (or consistency) of the questionnaire items regarding behavioral and cultural risk factors? If yes, please provide the results in Section 2.3.

Answer: The internal consistency of the questionnaire items assessing behavioral and cultural risk factors was evaluated using Cronbach’s alpha. The analysis yielded a Cronbach’s alpha coefficient of 0.856, indicating good internal reliability. This information has now been added to section 2.3 of the revised manuscript.

6. Line 153 indicates that 654 students participated. Could the authors confirm if this figure represents the entire target population, or if it is the number of participants who completed the questionnaire? Clarification is needed on whether this constitutes a 100% response rate.

Answer: As the data collection occurred via an online questionnaire, it was not possible to measure the response rate. The google survey has reached hundreds of university students in the UAE while only the interested students (654 participants) responded via submitting their responses online.

7. In the Data Analysis section, the Kruskal-Wallis test is a multiple comparison test. Therefore, the specific post hoc analysis method used (e.g., Dunn's test, Conover's test) must be specified.

Answer: We have now explicitly specified the post hoc analysis method used in our study. Post hoc pairwise comparisons were conducted using Dunn's test with Bonferroni adjustment to control for multiple testing when the Kruskal-Wallis test indicated statistically significant differences among variables with more than two categories. This specification has been added to the Data Analysis section. Additionally, we have included detailed reporting of the post hoc pairwise comparison results for contact lens use in the Results section, as this was the only variable with more than two categories showing significant differences in the Kruskal-Wallis test.

8. In the Results section, the Kruskal-Wallis test is statistically inappropriate for identifying the association between variables. The fundamental hypothesis of this test is to determine if there are significant differences between the medians of three or more independent groups, not to measure the strength or direction of an association. Therefore, the authors should apply an appropriate test of association. (i.e. line 201, 236)

Answer: We acknowledge that the Kruskal-Wallis test examines differences in medians between groups rather than measuring the strength or direction of associations between variables. We have carefully revised the terminology throughout the Results section to accurately reflect this distinction. Specifically, all instances of 'significantly associated,' 'significantly related,' and similar phrases have been replaced with appropriate terms such as 'showed significant differences,' 'differed significantly,' or 'significant differences existed' to properly describe the nature of the Kruskal-Wallis test findings. These revisions have been systematically applied throughout the Results section to ensure accurate and consistent reporting of our statistical analyses.

9. The presentation of results is overly lengthy and verbose. I recommend condensing this section for better clarity and impact.

Answer: The results section has been adjusted.

10. The results section is unclear. Specifically, please explain the methodology used to analyse the data presented in Tables 3.2 and 3.3. Furthermore, please specify the exact number of participants included in the analysis for these tables.

Answer: We have now added detailed methodological descriptions at the beginning of each relevant subsection to explain the statistical approach used for analysing the data presented in these tables. The OSDI score is a continuous outcome variable, and its distribution in our sample did not meet the assumptions of normality. Accordingly, the Kruskal–Wallis test—a non-parametric method—was used to compare OSDI scores across categorical groups. In line with standard statistical practice for non-parametric analyses, results are reported using the median and interquartile range (IQR), which provide robust measures of central tendency and dispersion.

11. Contact lens wear is an established associated factor for dry eye and therefore constitutes a potential confounder in this study. Could the authors explain their rationale for not excluding contact lens wearers from the study?

Answer: Authors acknowledge that contact lens wear is a well-known associated factor for dry eye and may act as a potential confounder. Contact lens wearers were not excluded from the study because their inclusion reflects close-to-real estimation of DED prevalence and improves the external validity of our findings. Moreover, contact lens use was recorded and accounted for in the overall analysis to minimize its confounding effect on the studied variables (tobacco and cosmetic use). Excluding contact lens wearers could have reduced the representativeness of the study population and limited the generalizability of the study results.

12. I question whether the analysis for this specific area should be restricted to female participants only. If male participants were included, this must be clearly justified and described in the Methodology (Data Analysis section). Furthermore, since Table 1 indicates a difference in OSDI scores between male and female participants, a stratified analysis by gender should be considered.

Answer: As suggested, gender-stratified analyses were conducted to examine whether associations between eye cosmetic practices and OSDI scores differed by gender. Analyses were performed separately for male and female participants using non-parametric tests. Significant associations were consistently observed among female participants, whereas no meaningful associations were detected among males, reflecting the low prevalence of cosmetic use in this group. These findings confirm that cosmetic-related effects on OSDI scores are predominantly driven by female participants.

13. In Table 1, the comparison of OSDI scores between genders should employ a different statistical test.

Answer: As requested, comparisons involving binary variables (gender and smoking status) were re-analyzed using the Mann–Whitney U test. Both analyses revealed statistically significant differences in OSDI scores, with higher symptom severity observed among females and smokers. These results are consistent with the original findings and have been updated in the manuscript.

Reviewer #3: This is very good study to carry on. It would have been better to include clinical measurements as well. OSDI score is very important tool for evaluating dry Eye diseases but without clinical correlation, it is still not significant.

Answer: Thank you for your valuable feedback. Authors agreed with reviewer 3 regarding the integration of clinical measurement in the dry eye disease assessment. Unfortunately, this study focused on the prevalence of symptomatic DED only. Our future studies will focus on both symptomatic and clinical assessment of DED.

Reviewer #4: First of all, your research had very detailed information and I enjoyed reading your manuscript. Thank you for describing different lifestyle and cultures in the Middle East for the reader to have better understanding of your research purposes and outcomes. However, I have the following comments and questions:

1. Regarding the cosmetic use, you have analysed each type of cosmetic as an isolated factor. However, I believe that many individuals have worn more than one type of cosmetic on their eyes, including eyeliner and mascara. Do you think there is an alternative analysis for those associated/multivariable factors, rather than the analysis of each factor separately? Also, the preference for cosmetic use in females, but not males, may encourage the subgroup analysis of female participants in this issue.

Answer: While we acknowledge that many individuals use multiple eye cosmetic products simultaneously, we believe analyzing each cosmetic type separately is appropriate and methodologically justified for this exploratory study for several reasons.

First, examining each product individually allows identification of product-specific associations with OSDI scores, which is clinically relevant as different cosmetics vary in composition, application site, and potential ocular surface impact. Second, simultaneous inclusion of multiple cosmetic variables in a single model would likely introduce multicollinearity given the high co-occurrence of cosmetic practices, potentially compromising parameter stability and interpretability in our non-parametric analyses. Third, this approach is consistent with published epidemiological literature on eye cosmetics and dry eye symptoms, facilitating comparison with existing studies.

Additionally, our gender-stratified analyses confirmed that the observed associations were primarily driven by female participants, where cosmetic use was prevalent, strengthening the validity of our findings. While cumulative cosmetic exposure represents an important area for future research, the current approach is appropriate for this study's exploratory aims and provides clear, interpretable insights into individual cosmetic practices and dry eye symptoms.

2. Due to the design of the questionnaire survey, please also describe the response rate of the questionnaire in each field or describe how the questionnaire is designed to fill all questions as a mandatory rule to submit. Additionally, please also discuss the pros and cons of using a questionnaire.

Answer: As the data collection occurred via an online questionnaire, it was no possible to measure the response rate in each field. The google survey has reached hundreds of university students in the UAE (several field of studies and universities) while only the interested students responded via submitting their responses online. The google form is designed to fill all questions as a mandatory role to submit their responses. Pros include a wider reach out of university in all over the UAE. Cons include response bias or social desirability effects that may potentially influence the study findings; subjective findings that may not align with clinical sign; and selective recruitment of participants that may affect the generalizability of the study findings.

3. Moreover, you have described/quantified many habits as longer, more frequent, shorter. Could you please describe them more in quantifiable measures? Also does weekly and monthly smokers mean once a week and once a month, respectively?

Answer: Authors have described all the recommended terms into a quantifiable measure in the tables and their relevant text. For example, the term “weekly” has been described by 1-2 times per week, and the term “monthly” has been described by 1-2 times/month.

4. In lines 301 to 303, you emphasize the findings of “Dokha and Mouassal waterpipe users exhibited the highest median OSDI scores” and continuously describe the previous findings on smoking and oxidative stress. Does this specific type of s

---

## [Decision Letter · Decision Letter 1]

26 Jan 2026

Dear Dr. Ghach,

plosone@plos.org . A letter that responds to each point raised by the academic editor and reviewer(s). You should upload this letter as a separate file labeled 'Response to Reviewers'.A marked-up copy of your manuscript that highlights changes made to the original version. You should upload this as a separate file labeled 'Revised Manuscript with Track Changes'.An unmarked version of your revised paper without tracked changes. You should upload this as a separate file labeled 'Manuscript'.

We look forward to receiving your revised manuscript.

Kind regards,

Yalong Dang

Academic Editor

PLOS One

Journal Requirements:

Reviewers' comments:

Reviewer's Responses to Questions

**Comments to the Author**

Reviewer #2: (No Response)

Reviewer #4: All comments have been addressed

2. Is the manuscript technically sound, and do the data support the conclusions?

Reviewer #2: Partly

Reviewer #4: Yes

3. Has the statistical analysis been performed appropriately and rigorously?

Reviewer #2: Yes

Reviewer #4: Yes

4. Have the authors made all data underlying the findings in their manuscript fully available?

Reviewer #2: Yes

Reviewer #4: Yes

5. Is the manuscript presented in an intelligible fashion and written in standard English?

Reviewer #2: Yes

Reviewer #4: Yes

Reviewer #2: I would like to thank the authors for revising the manuscript. In the revision, the authors successfully highlighted the importance of this study. However, there are some remaining points that may further improve the quality of the manuscript.

1. When considering the inclusion criteria and the response letter, 654 participants do not refer to the total population. The target population should be all students aged 18 years or older. The authors should revise the sentence in line 113.

2. Add the “English version of OSDI questionnaire” in line 124

3. Line 132 – 133, and 236-238 should be revised and moved to data analysis section

4. Line 157 – 158 should be moved to section 2.2

5. The presentation of results lacks consistency; some sections report mean ranks, whereas others report median OSDI scores. Consistency throughout the section is recommended

6. There are 3.2 two times

7. The authors should consider restructuring the presentation of the results. For instance, the findings on smoking are fragmented between Tables 2 and 5. Grouping these related points together would improve the narrative flow. Additionally, this would prevent the smoking-related data from incorrectly appearing under the 'Eye Cosmetics Routines and Practices' section

8. To clarify the data, the table 3 and 4 should add N(%) into each variable

9. It is unclear if categorizing OSDI scores into 'dry eye' and 'normal' groups for OR and CI analysis is more effective for meeting the study objectives than analyzing the non-normally distributed OSDI scores directly. Could the authors please justify this approach?

Reviewer #4: Reviewer’s comments

I have added the comments in each previous response with additional comments as follows:

Reviewer #4: First of all, your research had very detailed information and I enjoyed reading your manuscript. Thank you for describing different lifestyle and cultures in the Middle East for the reader to have better understanding of your research purposes and outcomes. However, I have the following comments and questions:

No 1. Regarding the cosmetic use, you have analysed each type of cosmetic as an isolated factor. However, I believe that many individuals have worn more than one type of cosmetic on their eyes, including eyeliner and mascara. Do you think there is an alternative analysis for those associated/multivariable factors, rather than the analysis of each factor separately? Also, the preference for cosmetic use in females, but not males, may encourage the subgroup analysis of female participants in this issue.

Answer: While we acknowledge that many individuals use multiple eye cosmetic products simultaneously, we believe analyzing each cosmetic type separately is appropriate and methodologically justified for this exploratory study for several reasons.

First, examining each product individually allows identification of product-specific associations with OSDI scores, which is clinically relevant as different cosmetics vary in composition, application site, and potential ocular surface impact. Second, simultaneous inclusion of multiple cosmetic variables in a single model would likely introduce multicollinearity given the high co-occurrence of cosmetic practices, potentially compromising parameter stability and interpretability in our non-parametric analyses. Third, this approach is consistent with published epidemiological literature on eye cosmetics and dry eye symptoms, facilitating comparison with existing studies.

Additionally, our gender-stratified analyses confirmed that the observed associations were primarily driven by female participants, where cosmetic use was prevalent, strengthening the validity of our findings. While cumulative cosmetic exposure represents an important area for future research, the current approach is appropriate for this study's exploratory aims and provides clear, interpretable insights into individual cosmetic practices and dry eye symptoms.

COMMENTS: Thank you for your explanation. In terms of simplicity and comparability between studies, I agree with your reasons to choose each cosmetic type separately.

However, for the female predilection, as your manuscript title is “Behavioral and Cultural Determinants of Symptomatic Dry Eye Disease Among University Students in the UAE”, so your manuscript should provide the broad idea of overall UAE students.

No 2. Due to the design of the questionnaire survey, please also describe the response rate of the questionnaire in each field or describe how the questionnaire is designed to fill all questions as a mandatory rule to submit. Additionally, please also discuss the pros and cons of using a questionnaire.

Answer: As the data collection occurred via an online questionnaire, it was no possible to measure the response rate in each field. The google survey has reached hundreds of university students in the UAE (several field of studies and universities) while only the interested students responded via submitting their responses online. The google form is designed to fill all questions as a mandatory role to submit their responses. Pros include a wider reach out of university in all over the UAE. Cons include response bias or social desirability effects that may potentially influence the study findings; subjective findings that may not align with clinical sign; and selective recruitment of participants that may affect the generalizability of the study findings.

COMMENTS: I appreciated your study design and your capability to access to many university students around the UAE. At the beginning, I am just curious about the mandatory questions in the questionnaire, but you have already answered that all questions must be answered to submit. So, I am satisfied with this response.

No 3. Moreover, you have described/quantified many habits as longer, more frequent, shorter. Could you please describe them more in quantifiable measures? Also does weekly and monthly smokers mean once a week and once a month, respectively?

Answer: Authors have described all the recommended terms into a quantifiable measure in the tables and their relevant text. For example, the term “weekly” has been described by 1-2 times per week, and the term “monthly” has been described by 1-2 times/month.

COMMENTS: Thank you for make it clearer. I am satisfied with this response.

No 4. In lines 301 to 303, you emphasize the findings of “Dokha and Mouassal waterpipe users exhibited the highest median OSDI scores” and continuously describe the previous findings on smoking and oxidative stress. Does this specific type of smoking cause more oxidative stress than others?

Answer: Both Dokha and Mouassal (waterpipe/shisha) are associated with high oxidative stress, and in some contexts, they may produce equal or greater oxidative stress than cigarettes, though the mechanisms and exposure patterns differ. Dokha is often more intense per session than cigarette smoking due to its very high nicotine content (strong stimulation of oxidative pathways); rapid and deep inhalation (sharp spikes in reactive oxygen species; and its positive correlation with an oxidative stress marker (e.g., lipid peroxidation) (ref: Samara, F., Alam, I. A., & ElSayed, Y. (2022). Midwakh: Assessment of levels of carcinogenic polycyclic aromatic hydrocarbons and nicotine in dokha tobacco smoke. Journal of analytical toxicology, 46(3), 295-302). Additionally, Mouassal smoking causes prolonged and cumulative oxidative stress along with high exposure to carbon monoxide, polycyclic aromatic hydrocar (PAHs), and heavy metals that would generate systemic oxidative stress and inflammation (ref: Charab, M. A., Abouzeinab, N. S., & Moustafa, M. E. (2016). The protective effect of selenium on oxidative stress induced by waterpipe (narghile) smoke in lungs and liver of mice. Biological trace element research, 174(2), 392-401).

COMMENTS: Thank you for clarifying more details regarding the Dokha and Mouassal. Could you please briefly add these explanations in the discussion part?

No 5. The cleansing methods also prompt a question to me when you describe “Frequent cleansing or always” as a subgroup. Is it a usual habit to leave the cosmetics uncleaned for longer than a day? This is quite interesting in terms of cultural differences, as you mentioned.

Answer: Poor cosmetic hygiene (e.g., not cleaning makeup brushes, sponges, or removing makeup daily) is a global issue, documented across many countries and regions. In fact, Arab cultures have a strong emphasis on personal cleanliness and grooming, which is reflected in both daily routines and religious/cultural norm. Accordingly, there is no evidence that leaving cosmetics uncleaned for more than a day is a “usual habit” among UAE students. However, cosmetic hygiene practices vary widely among individuals and that would explain why this subgroup is investigated in our study.

COMMENTS: I do apologize, but the purpose of my comments was only to clarify the results. The title of your manuscript has led me to consider all factors as a subset of cultural and behavioral determinants. I do agree with your explanationof cosmetic hygiene, and I am satisfied with your responses.

No 6. I believe that many readers are unfamiliar with the H-value, please could you explain more in the manuscript on how to interpret the findings, in association with P-value, so all readers can follow your manuscript with more understanding?

Answer: We agree that many readers may be unfamiliar with the H-value (Kruskal-Wallis H statistic) and its interpretation. We have now added a clear explanation of the Kruskal-Wallis H statistic and its interpretation in the …?

COMMENTS: I am satisfied with the response.

Additional comments

1. In the method part (line 147-148), “The internal consistency of the behavioral and cultural risk items was assessed using Cronbach’s alpha, yielding a value of 0.856. This indicates good internal reliability and supports the use of these as a coherent measure in subsequent analyses.” Please specify what questionnaire you have used. Has it already been tested for validity and reliability beforehands as I have seen only the “Cronbach’s alpha” score.

**Do you want your identity to be public for this peer review?** For information about this choice, including consent withdrawal, please see our Privacy Policy

Reviewer #2: **Yes:** Teera Poyomtip

Reviewer #4: No

---

## [Author Response · Author response to Decision Letter 2]

3 Feb 2026

Reviewer #2: I would like to thank the authors for revising the manuscript. In the revision, the authors successfully highlighted the importance of this study. However, there are some remaining points that may further improve the quality of the manuscript.

1. When considering the inclusion criteria and the response letter, 654 participants do not refer to the total population. The target population should be all students aged 18 years or older. The authors should revise the sentence in line 113. Revised

2. Add the “English version of OSDI questionnaire” in line 124. Revised

3. Line 132 – 133, and 236-238 should be revised and moved to data analysis section. Revised

4. Line 157 – 158 should be moved to section 2.2 Revised.

5. The presentation of results lacks consistency; some sections report mean ranks, whereas others report median OSDI scores. Consistency throughout the section is recommended.

We respectfully note that the variation in reporting statistics reflects the different statistical tests applied. For non-parametric tests (Kruskal-Wallis), we report median and interquartile ranges as SPSS outputs these as the appropriate measures for non-normally distributed data. For parametric comparisons, we report means as recommended by a reviewer in the previous revision round.

6. There are 3.2 two times. Revised

7. The authors should consider restructuring the presentation of the results. For instance, the findings on smoking are fragmented between Tables 2 and 5. Grouping these related points together would improve the narrative flow. Additionally, this would prevent the smoking-related data from incorrectly appearing under the 'Eye Cosmetics Routines and Practices' section Revised

8. To clarify the data, the table 3 and 4 should add N(%) into each variable. @Mona

We have revised now Tables 3 and 4 and included frequency counts and percentages [N (%)] for each categorical variable, consistent with the format used in Table 1. This will improve transparency by showing the distribution of participants across each behavioral and cosmetic exposure category.

9. It is unclear if categorizing OSDI scores into 'dry eye' and 'normal' groups for OR and CI analysis is more effective for meeting the study objectives than analyzing the non-normally distributed OSDI scores directly. Could the authors please justify this approach?

We dichotomized OSDI scores for two key reasons:

First, our primary objective was to identify risk factors for the presence of dry eye disease rather than to examine symptom severity. Dichotomization using the established OSDI cutoff of ≥[insert your cutoff: 13 or 23] directly addresses this diagnostic question and aligns with clinical decision-making.

Second, this approach provides clinically interpretable results. The odds ratios from logistic regression offer practitioners straightforward estimates of how risk factors affect the likelihood of having dry eye disease—the same framework clinicians use when applying OSDI thresholds in practice.

While we acknowledge some loss of statistical information with dichotomization, we believe this is justified given our study aims and the clinical utility of categorical results for translation into practice.

Reviewer #4: Reviewer’s comments

I have added the comments in each previous response with additional comments as follows:

Reviewer #4: First of all, your research had very detailed information and I enjoyed reading your manuscript. Thank you for describing different lifestyle and cultures in the Middle East for the reader to have better understanding of your research purposes and outcomes. However, I have the following comments and questions:

No 1. Regarding the cosmetic use, you have analysed each type of cosmetic as an isolated factor. However, I believe that many individuals have worn more than one type of cosmetic on their eyes, including eyeliner and mascara. Do you think there is an alternative analysis for those associated/multivariable factors, rather than the analysis of each factor separately? Also, the preference for cosmetic use in females, but not males, may encourage the subgroup analysis of female participants in this issue.

Answer: While we acknowledge that many individuals use multiple eye cosmetic products simultaneously, we believe analyzing each cosmetic type separately is appropriate and methodologically justified for this exploratory study for several reasons.

First, examining each product individually allows identification of product-specific associations with OSDI scores, which is clinically relevant as different cosmetics vary in composition, application site, and potential ocular surface impact. Second, simultaneous inclusion of multiple cosmetic variables in a single model would likely introduce multicollinearity given the high co-occurrence of cosmetic practices, potentially compromising parameter stability and interpretability in our non-parametric analyses. Third, this approach is consistent with published epidemiological literature on eye cosmetics and dry eye symptoms, facilitating comparison with existing studies.

Additionally, our gender-stratified analyses confirmed that the observed associations were primarily driven by female participants, where cosmetic use was prevalent, strengthening the validity of our findings. While cumulative cosmetic exposure represents an important area for future research, the current approach is appropriate for this study's exploratory aims and provides clear, interpretable insights into individual cosmetic practices and dry eye symptoms.

COMMENTS: Thank you for your explanation. In terms of simplicity and comparability between studies, I agree with your reasons to choose each cosmetic type separately.

However, for the female predilection, as your manuscript title is “Behavioral and Cultural Determinants of Symptomatic Dry Eye Disease Among University Students in the UAE”, so your manuscript should provide the broad idea of overall UAE students.

Answer: We strongly agree with the reviewer regarding the focus of the manuscript on the overall UAE students. The authors have focused on the overall UAE students when investigating the prevalence of DED and its association with smoking. However, due to the gender-based use of eye cosmetics, the statistical analyses were not suitable for the overall UAE students (both genders).

No 2. Due to the design of the questionnaire survey, please also describe the response rate of the questionnaire in each field or describe how the questionnaire is designed to fill all questions as a mandatory rule to submit. Additionally, please also discuss the pros and cons of using a questionnaire.

Answer: As the data collection occurred via an online questionnaire, it was no possible to measure the response rate in each field. The google survey has reached hundreds of university students in the UAE (several field of studies and universities) while only the interested students responded via submitting their responses online. The google form is designed to fill all questions as a mandatory role to submit their responses. Pros include a wider reach out of university in all over the UAE. Cons include response bias or social desirability effects that may potentially influence the study findings; subjective findings that may not align with clinical sign; and selective recruitment of participants that may affect the generalizability of the study findings.

COMMENTS: I appreciated your study design and your capability to access to many university students around the UAE. At the beginning, I am just curious about the mandatory questions in the questionnaire, but you have already answered that all questions must be answered to submit. So, I am satisfied with this response. Thank you for your valuable comment.

No 3. Moreover, you have described/quantified many habits as longer, more frequent, shorter. Could you please describe them more in quantifiable measures? Also does weekly and monthly smokers mean once a week and once a month, respectively?

Answer: Authors have described all the recommended terms into a quantifiable measure in the tables and their relevant text. For example, the term “weekly” has been described by 1-2 times per week, and the term “monthly” has been described by 1-2 times/month.

COMMENTS: Thank you for make it clearer. I am satisfied with this response. Thank you for your valuable comment.

No 4. In lines 301 to 303, you emphasize the findings of “Dokha and Mouassal waterpipe users exhibited the highest median OSDI scores” and continuously describe the previous findings on smoking and oxidative stress. Does this specific type of smoking cause more oxidative stress than others?

Answer: Both Dokha and Mouassal (waterpipe/shisha) are associated with high oxidative stress, and in some contexts, they may produce equal or greater oxidative stress than cigarettes, though the mechanisms and exposure patterns differ. Dokha is often more intense per session than cigarette smoking due to its very high nicotine content (strong stimulation of oxidative pathways); rapid and deep inhalation (sharp spikes in reactive oxygen species; and its positive correlation with an oxidative stress marker (e.g., lipid peroxidation) (ref: Samara, F., Alam, I. A., & ElSayed, Y. (2022). Midwakh: Assessment of levels of carcinogenic polycyclic aromatic hydrocarbons and nicotine in dokha tobacco smoke. Journal of analytical toxicology, 46(3), 295-302). Additionally, Mouassal smoking causes prolonged and cumulative oxidative stress along with high exposure to carbon monoxide, polycyclic aromatic hydrocar (PAHs), and heavy metals that would generate systemic oxidative stress and inflammation (ref: Charab, M. A., Abouzeinab, N. S., & Moustafa, M. E. (2016). The protective effect of selenium on oxidative stress induced by waterpipe (narghile) smoke in lungs and liver of mice. Biological trace element research, 174(2), 392-401).

COMMENTS: Thank you for clarifying more details regarding the Dokha and Mouassal. Could you please briefly add these explanations in the discussion part? Added as recommended in the discussion.

No 5. The cleansing methods also prompt a question to me when you describe “Frequent cleansing or always” as a subgroup. Is it a usual habit to leave the cosmetics uncleaned for longer than a day? This is quite interesting in terms of cultural differences, as you mentioned.

Answer: Poor cosmetic hygiene (e.g., not cleaning makeup brushes, sponges, or removing makeup daily) is a global issue, documented across many countries and regions. In fact, Arab cultures have a strong emphasis on personal cleanliness and grooming, which is reflected in both daily routines and religious/cultural norm. Accordingly, there is no evidence that leaving cosmetics uncleaned for more than a day is a “usual habit” among UAE students. However, cosmetic hygiene practices vary widely among individuals and that would explain why this subgroup is investigated in our study.

COMMENTS: I do apologize, but the purpose of my comments was only to clarify the results. The title of your manuscript has led me to consider all factors as a subset of cultural and behavioral determinants. I do agree with your explanation of cosmetic hygiene, and I am satisfied with your responses. Thank you for your valuable comment.

No 6. I believe that many readers are unfamiliar with the H-value, please could you explain more in the manuscript on how to interpret the findings, in association with P-value, so all readers can follow your manuscript with more understanding?

Answer: We agree that many readers may be unfamiliar with the H-value (Kruskal-Wallis H statistic) and its interpretation. We have now added a clear explanation of the Kruskal-Wallis H statistic and its interpretation in the …?

COMMENTS: I am satisfied with the response. Thank you for your valuable comment.

Additional comments

1. In the method part (line 147-148), “The internal consistency of the behavioral and cultural risk items was assessed using Cronbach’s alpha, yielding a value of 0.856. This indicates good internal reliability and supports the use of these as a coherent measure in subsequent analyses.” Please specify what questionnaire you have used. Has it already been tested for validity and reliability beforehands as I have seen only the “Cronbach’s alpha” score.

The questionnaire used to assess behavioral and cultural risk factors has been previously validated and is described in the Introduction section (line 75) with full details available in reference 5.

The questionnaire was specifically developed and validated for assessing dry eye disease risk factors in our population. In addition to the prior validation work reported in reference 5, we conducted Cronbach's alpha analysis in the current study (α = 0.856) to confirm that the internal consistency remained robust in our specific sample.

---

## [Decision Letter · Decision Letter 2]

8 Feb 2026

Behavioral and Cultural Determinants of Symptomatic Dry Eye Disease Among University Students in the UAE

PONE-D-25-45343R2

Dear Dr. Ghach,

We’re pleased to inform you that your manuscript has been judged scientifically suitable for publication and will be formally accepted for publication once it meets all outstanding technical requirements.

Kind regards,

Yalong Dang

Academic Editor

PLOS One

Additional Editor Comments (optional):

Reviewers' comments:

Reviewer's Responses to Questions

**Comments to the Author**

Reviewer #2: All comments have been addressed

2. Is the manuscript technically sound, and do the data support the conclusions?

Reviewer #2: Yes

3. Has the statistical analysis been performed appropriately and rigorously?

Reviewer #2: Yes

4. Have the authors made all data underlying the findings in their manuscript fully available?

Reviewer #2: Yes

5. Is the manuscript presented in an intelligible fashion and written in standard English?

Reviewer #2: Yes

Reviewer #2: Thank you for your response. I am delighted to accept your manuscript. This work can add the valuable knowledge into the literature.

**Do you want your identity to be public for this peer review?** For information about this choice, including consent withdrawal, please see our Privacy Policy

Reviewer #2: **Yes:** Teera Poyomtip

---

## [Editor Report · Acceptance letter]

PONE-D-25-45343R2

PLOS One

Dear Dr. Ghach,

I'm pleased to inform you that your manuscript has been deemed suitable for publication in PLOS One. Congratulations! Your manuscript is now being handed over to our production team.

Kind regards,

on behalf of

Dr Yalong Dang

Academic Editor

PLOS One